# Measuring the Uptake of Growth Monitoring and Nutrition Promotion among under-5 Children: Findings from the Rwanda Population-Based Study

**DOI:** 10.3390/children9111758

**Published:** 2022-11-16

**Authors:** Michael Ekholuenetale, Amadou Barrow, Anthony Ike Wegbom, Amit Arora

**Affiliations:** 1Department of Epidemiology and Medical Statistics, Faculty of Public Health, College of Medicine, University of Ibadan, Ibadan 200284, Nigeria; 2Department of Public & Environmental Health, School of Medicine & Allied Health Sciences, University of The Gambia, Kanifing 3530, The Gambia; 3Department of Public Health Sciences, College of Medical Sciences, Rivers State University, Port Harcourt 500101, Nigeria; 4Translational Health Research Institute, Western Sydney University, Campbelltown, NSW 2560, Australia; 5School of Health Sciences, Western Sydney University, Penrith, NSW 2751, Australia; 6Health Equity Laboratory, Campbelltown, NSW 2560, Australia; 7Discipline of Child and Adolescent Health, The Children’s Hospital at Westmead Clinical School, Faculty of Medicine and Health, The University of Sydney, Westmead, NSW 2145, Australia; 8Oral Health Services, Sydney Local Health District and Sydney Dental Hospital, NSW Health, Surry Hills, NSW 2010, Australia

**Keywords:** inequality, growth monitoring, child health, nutrition promotion, malnutrition, undernutrition, first 2000 days

## Abstract

Regular growth monitoring can be used to evaluate the nutritional and physical health of children. Ample evaluation of the reach and quality of nutrition interventions is necessary to increase their effectiveness, but there is little research on improving coverage measurement. The aim of this study was to explore the coverage of growth monitoring, nutrition promotion, and associated factors by Rwandan caregivers of children under the age of five. Data from 2019–2020 Rwanda Demographic and Health Survey with a total of 8092 children under the age of five were used for this study. Prevalence of growth monitoring and nutrition promotion were reported and the factors influencing this were evaluated using multivariable logistic regression model. The prevalence of growth monitoring and nutrition promotion among under-5 children was 33.0% (95%CI: 30.6–35.6%). Older children, caregivers who were native residents, those with a health insurance, in a marital relationship, employed, and residing in rural areas had higher odds to participate in growth monitoring and nutrition promotion compared to their counterparts. Rwanda has a low rate of coverage for growth monitoring and nutrition promotion among children <5 and public health nutrition interventions should prioritize nutritional counseling as well as the availability of growth monitoring and promotion services.

## 1. Background

Child malnutrition is a global public health issue. Inadequate or excessive nutrient intake, an imbalance of vital nutrients, or poor nutrient utilization are all considered forms of malnutrition. Undernutrition, overweight, and obesity all contribute to the double burden of malnutrition. About 45% of deaths in children under the age of five are attributed to undernutrition [1]. The resource-constrained settings account for the majority of sub-optimal feeding among children. Moreover, the rates of childhood obesity and overweight are also increasing in resource-poor countries. Globally, approximately 45.4 million wasted, 38.9 million overweight, and 149.2 million under-5 children had stunted growth by 2020 [2]. All regions, with the exception of Africa, are seeing reduction in the burden of stunted children. Of all the progress made in various indicators of child health, the reduction in stunting remains the greatest achievement made in about two-thirds of countries [2]. On the other hand, regarding overweight, roughly half of all countries have not made any progress or have seen a worsening situation [2].

According to the conceptual framework of the United Nations Children’s Fund (UNICEF), nutrition, health, and psychosocial stimulation are the essential elements required to enhance children’s quality of life [3]. To improve health and development, proper feeding practices must be promoted [4]. The Sustainable Development Goals (SDGs), particularly those that aim to end poverty in all of its forms worldwide (SDG 1), end hunger, achieve food security and improve nutrition, and promote sustainable agriculture (SDG 2), as well as ensure healthy lives and promote wellbeing for all people of all ages (SDG 3), must be accomplished in large part by reducing malnutrition among children [5,6]. Many countries have endorsed global targets to lower stunting (chronic undernutrition) by 40% by 2025 and to lower and maintain the prevalence of wasting (acute undernutrition) in children under five years old to less than 5%. This is due to the serious consequences of childhood malnutrition [7]. Considering the global initiatives to enhance infant and child feeding practices through the International Code of Marketing of Breast milk Substitutes, the promotion of proper nutrition, including breastfeeding [8], the Global Strategy for Infant and Young Child Feeding [9], and the baby-friendly hospital initiative (BFHI) [10], are crucial part of a child’s growth mechanisms.

The prevalence of stunting among children under the age of five in Rwanda decreased from 47.4% in 2000 to 38.3% in 2015 [11]. The high rate of undernutrition among children has reportedly been linked to a number of factors. For instance, a study performed in Rwanda found that a high rate of stunting was linked to factors such as being a male, having a low birth weight, living in the lowest household wealth quintile, having a mother who smoked, and having a mother with limited education [11].

Malnutrition may be more prevalent in Rwanda as a result of the severe political and economic crisis that followed the 1994 genocide [12]. Since then, Rwanda has implemented a number of progressive measures outlined in its Vision 2020 plan to promote economic recovery [13]. As a result, the population’s health has significantly improved across a variety of population health parameters [14]. For instance, there has been a decline in infant and under-5 mortality and the vaccination rates have improved significantly between 2000 and 2015 [11]. This development could be related to extensive reforms and innovations created with community involvement to strengthen the health system, including implementing the community-based health insurance plan (known as *Mutuelles de Santé*) to improve financial access to care; creating a strong network of community health workers to provide care at the village level; and creating performance-based financing programs to increase the quality of healthcare services [15,16,17].

Malnutrition continues to be a significant burden for Rwandan children, despite the country’s improvements in health outcomes, with the rate of stunting remaining largely stable [18,19]. In Rwanda, the prevalence of stunting increased from 16.2% among children under the age of six months and reached its peak (40.4%) among children aged 24 to 35 months [19]. This represents the staggering occurrence of malnutrition in the first 1000 days of life [19]. To develop more effective programs and policies to address childhood malnutrition, it will be helpful to have a better understanding of the coverage and factors linked to child growth monitoring and nutrition promotion in Rwanda. This study was motivated by the fact that the persistent burden of malnutrition conflicts with the overall improvements in child health outcomes. Therefore, this study aims to explore the coverage of routine growth monitoring, nutrition promotion, and associated factors were utilized by Rwandan caregivers of children under the age of five.

## 2. Methods

### 2.1. Data Source

For this study, data from children questionnaires in the 2019–2020 Rwanda Demographic and Health Survey (RDHS) were extracted. A total sample of 8092 under-5 children were examined. The 2019–2020 RDHS was the sixth round following the 1992, 2000, 2005, 2010, and 2014–2015 surveys. The survey was carried out by the National Institute of Statistics of Rwanda, supported by Ministry of Health and Inner City Fund (ICF). The survey was conducted November 2019 through July 2020. Because of the impact of lockdown following the coronavirus pandemic in 2020, data collection was halted for about three months (March–June) [19]. The RDHS gathered information on factors, such as fertility rates and preferences, use of contraception, maternal and child health, rates of mortality, nutrition, knowledge of HIV/AIDS, and STIs, among others relevant to monitor population health [19]. A previous study has reported the methodology of RDHS [20].

### 2.2. Sampling Design

A two-stage stratified cluster sampling was adopted in the 2019–2020 RDHS, with the goal of allowing important indicators for the country to be estimated, while considering urban versus rural residence, geographical region (5 divisions) and districts (30 in Rwanda) for selected metrics. The National Institute of Statistics, RDHS’s implementing agency, provided a sampling frame of enumeration areas (EAs) for the entire country. EA is a community established for the 2012 Rwanda Population and Housing Census used as census counting unit. In the first step, sample clusters made up of EAs designated for the 2012 Rwanda Population and Housing Census were chosen. A total of 500 clusters were chosen, with 388 in rural and 112 in urban residence. Systematic household sampling was done in the second phase. From June to August 2019, a household listing was conducted in all chosen EAs, and households included in the survey were randomly chosen. Each cluster had approximately 26 households, thereby resulting in 13,000 households across the country.

### 2.3. Selection and Measurement of Variables

#### Outcome

The dependent variable for this study was measured in binary form and coded as “1” if “yes” was answered to the question: “Participated in monthly growth monitoring and nutrition promotion sessions” and “0” if otherwise.

### 2.4. Explanatory Variables

Mother’s age (years): 15–24, 25–34, 35+; family motility: <5 years versus 5+ years; mother’s education: no formal education, primary and secondary+; read newspaper: no versus yes; listen to radio: no versus yes; watch tv: no versus yes; covered by health insurance: no versus yes; mother’s marital status: never in union, currently in union/living with a man, and formerly in union; currently pregnant: no or unsure versus yes; currently breastfeeding: no versus yes; mother’s employment status: not employed versus employed; child’s age (months): 0–11, 12–23, 24–35, 36–47, and 48–59; sex of child: male versus female; preceding birth interval: <2 years, 2–4 year, 5+ years, and first born; place of delivery: health facility versus home; birthweight: low (<2.5 kg) versus normal (≥2.5 kg); sex of household head: male versus female; household wealth quintiles: poorest, poorer, middle, richer, and richest; residential status: urban versus rural; geographical region: Kigali, south, west, north, and east; household size: 1–4, 5–6, and 7+.

### 2.5. Ethical Consideration

A secondary dataset available in the public domain with identifier information removed, was analyzed for this study. RDHS followed a standard ethical procedure to obtain informed consent from respondents. There was no need for additional participants’ consent, as the authors obtained permission to use the dataset for the purpose of this study. The information regarding DHS ethical standards can be found here: http://goo.gl/ny8T6X (accessed on 20 September 2022).

### 2.6. Statistical Analysis

To compute the estimates of growth monitoring and nutrition promotion among under-5 children, Stata ‘svy’ module was used in adjusting for clustering, stratification, and sampling weights. Univariable and multivariable logistic regression analyses were performed to examine the potential predictors of growth monitoring and nutrition promotion among under-5 children. Variables that were significant at *p* < 0.2 in the univariable model were then included in the multivariable model. The adjusted odds ratios (AORs) with a 95% confidence interval (95%CI) are reported in the results. In the absence of multicollinearity, significant variables from the Chi-square test were retained in the logit model. Stata Version 16 was used for data analyses.

## 3. Results

The weighted prevalence of growth monitoring and nutrition promotion among under-5 children was 33.0% (95%CI: 30.6–35.6%). By implication, approximately two-thirds of under-5 children in Rwanda did not participate in growth monitoring and nutrition promotion services in 2019–2020.

Figure 1 shows the main reason for not participating in growth monitoring and nutrition promotion for under-5 children in Rwanda. Prominently, more than half (55.3%) of the respondents were not aware about under-5 growth monitoring and nutrition promotion sessions. In addition, the effect of COVID-19 (19.9%) and not having time to attend (15.1%) were the main reasons for not participating in growth monitoring and nutrition promotion among under-5 children in Rwanda.

Table 1 shows the weighted prevalence of growth monitoring and nutrition promotion across respondents’ characteristics. Approximately 38.3% of under-5 children from older mothers (35+ years) reported participating in growth monitoring and nutrition promotion. The native under-5 children reported 38.4% coverage of growth monitoring and nutrition promotion. About 44.8% of children from mothers with no formal education and about one-third of mothers that have health insurance, who are currently in a union, and who are employed reported participating in growth monitoring and nutrition promotion. The children from poor, uneducated, and rural residence reported higher prevalence of growth monitoring and nutrition promotion.

Table 2 shows the factors associated with under-5 growth monitoring and nutrition promotion. The native residents had 33% higher odds of participating in growth monitoring and nutrition promotion, when compared with families who have stayed <5 years in the location of residence (OR = 1.33; 95%CI: 1.14–1.56). Those covered by health insurance had 71% higher odds of participating in growth monitoring and nutrition promotion, when compared with those not covered by health insurance (OR = 1.71; 95%CI: 1.42–2.07). Married women had 1.66 times higher odds to present their children for growth monitoring and nutrition promotion, when compared with single women (OR = 1.66; 95%CI: 1.21–2.26). The under-5 children from caregivers who are currently employed had 17% higher odds to participate in growth monitoring and nutrition promotion, when compared with the unemployed (OR = 1.17; 95%CI: 1.01–1.36). The odds of participating in growth monitoring and nutrition promotion was positively associated with the age of children. Rural children had 1.79 times higher odds of participating in growth monitoring and nutrition promotion, when compared with those with urban residence (OR = 1.79; 95%CI: 1.44–2.23). The geographical region was significantly associated with participating in growth monitoring and nutrition promotion.

## 4. Discussion

The prevalence of uptake of growth monitoring and nutrition promotion services was 33% among under-5 children in Rwanda. This prevalence is lower than studies conducted in northwest Ethiopia (39%), Accra (64%), and Lawar, Ghana (60%) [21,22,23]. Additionally, this study’s results were lower than those from Ethiopia’s southern region (56%), rural Rwanda (90%), and northern Ethiopia (50%) [24,25,26]. A previous study among under-5 children showed a yearly and steady increase in the use of community-based growth-monitoring services from 53% in 2004 to 80% in 2008 [26]. There could be various reasons for the discrepancy, including differences in the study design, demographics of respondents, and study population. However, uptake of growth monitoring in Rwanda was implemented for under-5 populations, while Ethiopians between 0–23 months were targeted for growth monitoring [23]. Institution-based research with a relatively limited number of participants and a focus on immunization and unwell children have also been common in the past. This result indicates a greater demand for growth monitoring and promotion services.

The main reason why many mothers did not present their children for growth monitoring and nutrition promotion was due to a lack of awareness. According to a previous study [27], the majority of mothers were aware of the importance of routine height and weight checks, and health professionals also appeared knowledgeable about growth monitoring in accordance with global best practices. However, the study reported a lack of mothers’ knowledge regarding child feeding and a lack of essential resources to store and/or purchase nutritious food [27]. In general, if mothers are unaware of the importance of growth monitoring and nutrition promotion, growth monitoring is unlikely to succeed. Other prominent reasons reported were disruption of growth monitoring and health promotion sessions due to COVID-19 and caregivers not having the time to take their children to uptake the services. Poor nutrition among children has a major adverse effect in growth and cognitive development [28]. Hence, early detection is necessary to prevent its occurrence. Several factors have been reported to be responsible for hidden hunger among children, including household socioeconomic status [29]. Therefore, awareness creation or behavior change communication must target all households and the hard-to-reach communities.

The uptake of growth monitoring and nutrition promotion can be affected by various factors. The years lived in the current location, rural residence, and geographical region were significantly associated with the uptake of growth monitoring and nutrition promotion. The children from families who had lived more than 5 years in their residence (native residents) had higher uptake of growth monitoring and nutrition promotion services compared to those within 5 years of their stay in their residence. The native residents could probably be more aware of existing health services, including growth monitoring, than the non-natives. It is also possible that the native residents have better geographical access to service centers than the non-natives. Such differences in information and geographical access could be attributed to disparities in the uptake of growth monitoring among under-5 children. The rural residents and those from the southern, western, and northern geographical zones of Rwanda had higher odds of uptake of growth monitoring and nutrition promotion services, compared to those from urban and Kigali region. These findings were consistent with previous studies done in Rwanda [26,30]. In this study, about 79.0% of participants are rural residents. It is not unlikely that such representation could be linked to higher report of growth monitoring coverage among the rural residents. In addition, it is possible that the awareness campaign for growth monitoring service uptake may be more effective in the rural residence. It is possible that respondents within the same location, for example individuals with urban residence or those residing in certain geographical zones, may not be aware about the service or are prevented from participating due to COVID-19.

Children that were covered by health insurance and those from mothers who were employed had higher odds in the uptake of growth monitoring and nutrition promotion services. This is bolstered by findings of a Rwandan study that revealed increased enrollment and utilization into Rwandan community-based health insurance schemes among female-headed households compared to their male counterparts [31,32]. Mothers who are employed are most likely educated or have the ability to read and understand the importance of health promotion, enabling them to understand child health and growth needs. The children from unemployed mothers would have a low uptake of child health services and are prone to malnutrition, as highlighted in the national nutrition analysis [33].

The children from mothers who are currently married or in union had increased odds of utilizing growth monitoring and nutrition promotion services, as supported in similar studies in South Asia, Ethiopia, and Ghana [34,35,36]. The age of children was positively associated with the uptake of growth monitoring and nutrition promotion services. This is supported by the studies conducted in Kenya [8] and Ghana [9,10], which found a negative correlation between child’s age and uptake of growth monitoring and nutrition promotion. These could be attributed to the fact that growth monitoring and nutrition promotion services are delivered along with integrated routine child health services, including vaccination, as in the case of most African countries [35]. These allow children to uptake growth monitoring and nutrition promotion services along with immunization services. In a previous study, the age of children was significantly associated with the utilization of growth monitoring and promotion [35]. Thus, the findings of this study are consistent with the previous report on child’s age in relation to uptake of growth monitoring and nutrition promotion services in Rwanda.

### Strengths and Limitations

The use of a recently conducted household survey to gather nationally representative, high-quality data was a significant strength of this study. The reliability of the study’s findings is further ensured by appropriate statistical adjustments for the survey designs. This study’s primary outcome variable was based on self-report, which is susceptible to social desirability bias and recall effects. As a result, the coverage of growth monitoring and nutrition promotion may have been over- or under-estimated. The household income and expenditure data, which are the conventional metrics for calculating wealth, were not collected from the population-based survey. The asset-based wealth index used in this study is only a proxy indicator of household economic status and does not always yield results that are comparable to those obtained from direct income and expenditure measurements when such data are available or can be accurately collected. A significant flaw in the DHS data was the absence of pertinent variables, such as caregivers’ behavior regarding children’s health. Moreover, we were unable to evaluate the extent of availability of growth monitoring and health promotion sessions, which can also impact attendance. This was due to the use of secondary data which did not capture information on the availability of growth monitoring and health promotion sessions. As a result of the study’s cross-sectional design, only associations—not causality—can be deduced.

## 5. Conclusions

Children in Rwanda receive insufficient nutrition education and growth monitoring. The findings of this study show that only one-third of under-5 children participated in growth monitoring and nutrition promotion. Older children, caregivers who were native residents, those with a health insurance, and mothers who were in a marital union, employed, and residing in rural areas had higher odds to participate in growth monitoring and nutrition promotion compared to their counterparts. Nutritional interventions should, therefore, emphasize nutritional counseling and the availability of services for growth monitoring and promotion.

## Figures and Tables

**Figure 1 children-09-01758-f001:**
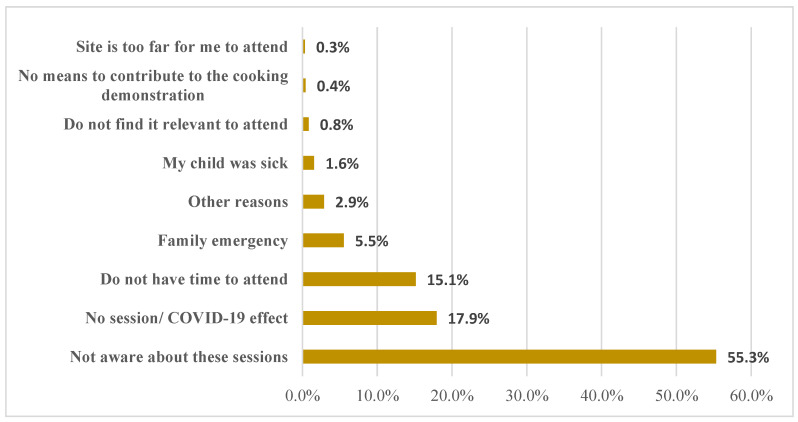
Main reasons for not participating in growth monitoring and nutrition promotion for under-5 children in Rwanda.

**Table 1 children-09-01758-t001:** Prevalence of under-5 growth monitoring and nutrition promotion (*n* = 8092).

Variable	*n* (%)	Prevalence of under-5 Growth Monitoring and Nutrition Promotion (95% Confidence Interval)	*p*
**Mother’s age (years)**			<0.001 *
15–24	1287 (15.9)	27.8 (24.9–30.8)	
25–34	3949 (48.8)	33.9 (32.0–35.9)	
35+	2856 (35.3)	38.3 (35.8–40.8)	
**Family motility**			<0.001 *
<5 years	2791 (34.5)	26.9 (24.8–29.1)	
5+ years (native)	5301 (65.5)	38.4 (36.6–40.2)	
**Mother’s education**			<0.001 *
No formal education	918 (11.3)	44.8 (40.5–49.2)	
Primary	5277 (65.2)	35.3 (33.6–37.1)	
Secondary+	1897 (23.4)	26.5 (24.0–29.1)	
**Mother read newspaper**			0.605
No	6528 (80.7)	33.9 (32.4–35.5)	
Yes	1564 (19.3)	34.8 (31.8–8.0)	
**Mother listen to radio**			0.150
No	1935 (23.9)	32.2 (29.5–35.1)	
Yes	6157 (76.1)	34.7 (33.1–36.3)	
**Mother watch tv**			0.395
No	4946 (61.1)	34.6 (32.8–36.4)	
Yes	3146 (38.9)	33.3 (31.2–35.6)	
**Covered by health insurance**			<0.001 *
No	1523 (18.8)	26.7 (23.7 to 29.9)	
Yes	6569 (81.2)	35.7 (34.2 to 37.2)	
**Mother’s marital status**			<0.001 *
Never in union	719 (8.9)	22.1 (18.1–26.6)	
Currently in union/living with a man	6734 (83.2)	35.6 (34.1–37.1)	
Formerly in union	639 (7.9)	29.3 (24.2–34.9)	
**Currently pregnant**			0.295
No or unsure	7568 (93.5)	34.0 (32.6–35.4)	
Yes	524 (6.5)	38.3 (30.5–46.9)	
**Currently breastfeeding**			0.335
No	2967 (36.7)	31.2 (25.5–37.4)	
Yes	5125 (63.3)	34.3 (32.9–35.7)	
**Mother’s employment status**			<0.001 *
Not employed	2009 (24.8)	29.6 (27.2–32.2)	
Employed	6083 (75.2)	35.9 (34.2–37.5)	
**Child’s age (months)**			<0.001 *
0–11	1618 (20.0)	29.1 (26.9–31.4)	
12–23	1633 (20.2)	36.4 (34.1–38.8)	
24–35	1643 (20.3)	35.7 (30.3–41.5)	
36–47	1633 (20.2)	35.3 (31.4–39.3)	
48–59	1565 (19.3)	39.6 (35.7–43.7)	
**Sex of child**			0.205
Male	4095 (50.6)	33.2 (31.3–35.2)	
Female	3997 (49.4)	35.0 (33.1–37.0)	
**Preceding birth interval**			<0.001 *
<2 years	867 (10.7)	34.9 (30.9–39.2)	
2–4 year	3618 (44.7)	38.1 (36.0–40.2)	
5+ years	1527 (18.9)	34.3 (31.0–37.7)	
First born	2080 (25.7)	27.2 (24.9–30.0)	
**Place of delivery**			0.844
Health facility	7663 (94.7)	34.1 (32.7–35.6)	
Home	429 (5.3)	33.4 (27.5–40.0)	
**Birthweight**			0.333
Low (<2.5 kg)	541 (7.1)	31.7 (26.4–37.4)	
Normal (≥2.5 kg)	7118 (92.9)	34.5 (33.1–36.0)	
**Sex of household head**			<0.001 *
Male	6257 (77.3)	35.4 (33.8–36.9)	
Female	1835 (22.7)	29.2 (26.3–32.2)	
**Household wealth quintiles**			<0.001 *
Poorest	2007 (24.8)	37.3 (34.5–40.1)	
Poorer	1569 (19.4)	39.2 (36.1–42.3)	
Middle	1518 (18.8)	36.9 (33.7–40.3)	
Richer	1510 (18.7)	33.0 (29.9–36.2)	
Richest	1488 (18.4)	22.1 (19.3–25.1)	
**Residential status**			<0.001 *
Urban	1702 (21.0)	19.9 (17.4–22.7)	
Rural	6390 (79.0)	37.5 (36.0–39.1)	
**Geographical region**			<0.001 *
Kigali	948 (11.7)	14.7 (12.0–18.0)	
South	1853 (22.9)	40.8 (37.8–43.8)	
West	2069 (25.6)	46.6 (43.8–49.3)	
North	1282 (15.8)	37.3 (33.7–41.0)	
East	1940 (24.0)	20.4 (18.1–23.0)	
**Household size**			0.107
1–4	3084 (38.1)	32.4 (30.1–34.7)	
5–6	3154 (39.0)	35.8 (33.6–38.0)	
7+	1854 (22.9)	33.8 (31.1–36.6)	

* Significant at *p* < 0.05; *p* value is for cross-tabulated significance estimation in the uptake or non-uptake of under-5 growth monitoring and nutrition promotion across the study factors (sociodemographics, among others).

**Table 2 children-09-01758-t002:** Factors associated with under-5 growth monitoring and nutrition promotion.

Variable	Adjusted Odds Ratio (95% Confidence Interval)	*p*
**Mother’s age (years)**		
15–24	RC	
25–34	0.95 (0.77–1.18)	0.659
35+	0.96 (0.75–1.23)	0.743
**Family motility**		
<5 years	RC	
5+ years (native)	1.33 (1.14–1.56)	<0.001 *
**Mother’s education**		
No formal education	RC	
Primary	0.76 (0.62–0.94)	0.010 *
Secondary+	0.83 (0.63–1.09)	0.176
**Covered by health insurance**		
No	RC	
Yes	1.71 (1.42–2.07)	<0.001 *
**Mother’s marital status**		
Never in union	RC	
Currently in union/living with a man	1.66 (1.21–2.26)	0.001 *
Formerly in union	1.34 (0.91–1.98)	0.144
**Mother’s employment status**		
Not employed	RC	
Employed	1.17 (1.01–1.36)	0.037 *
**Child’s age (months)**		
0–11	RC	
12–23	1.44 (1.23–1.69)	<0.001 *
24–35	1.34 (1.01–1.78)	0.043 *
36–47	1.30 (1.04–1.62)	0.019 *
48–59	1.45 (1.17–1.81)	0.001 *
**Preceding birth interval**		
<2 years	RC	
2–4 year	1.06 (0.85–1.32)	0.600
5+ years	0.99 (0.77–1.28)	0.934
First born	0.83 (0.64–1.07)	0.159
**Sex of household head**		
Male	RC	
Female	0.96 (0.79–1.16)	0.655
**Household wealth quintiles**		
Poorest	RC	
Poorer	1.02 (0.84–0.23)	0.865
Middle	0.96 (0.79–1.18)	0.706
Richer	0.99 (0.80–1.22)	0.896
Richest	0.74 (0.56–0.96)	0.023
**Residential status**		
Urban	RC	
Rural	1.79 (1.44–2.23)	<0.001 *
**Geographical region**		
Kigali	RC	
South	2.46 (1.83–3.30)	<0.001 *
West	3.17 (2.39–4.22)	<0.001 *
North	2.03 (1.49–2.78)	<0.001 *
East	0.89 (0.66–1.22)	0.478

RC, reference category; * significant at *p* < 0.05; *p* value is for significance testing in the odds of a factor, relative to the reference category in the uptake of under-5 growth monitoring and nutrition promotion.

## Data Availability

Data for this study were obtained from the National Demographic and Health Surveys (DHS) of the studied African countries, which can be found at http://dhsprogram.com/data/available-datasets.cfm (accessed on 20 September 2022).

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
