# Peer review of "Measuring the Uptake of Growth Monitoring and Nutrition Promotion among under-5 Children: Findings from the Rwanda Population-Based Study"

_children, 2022, doi:10.3390/children9111758_

Round 1

Reviewer 1 Report

The research was very interesting and brought forth a real and important problem.  The first step is to have appropriate data about the problem and the authors took care of that.  Also, it was very interesting the contrast between other similar studies.  This gives us the opportunity to make more studies and corroborate previous methodologies, to obtain the best data possible.  Another thing that I find interesting was the fact that the authors collected data on the reasons mothers did not took their children for monitoring.  This gives us information on what to improve to have more data and to give information to the participants.

Author Response

The research was very interesting and brought forth a real and important problem.  The first step is to have appropriate data about the problem and the authors took care of that.  Also, it was very interesting the contrast between other similar studies.  This gives us the opportunity to make more studies and corroborate previous methodologies, to obtain the best data possible.  Another thing that I find interesting was the fact that the authors collected data on the reasons mothers did not took their children for monitoring.  This gives us information on what to improve to have more data and to give information to the participants.

Response: Thank you very much for taking your time to read through our manuscript and providing us with very helpful comments.

Reviewer 2 Report

  • Increasing coverage of growth monitoring and nutrition promotion among under-5 children can be a tool to improve nutritional and physical health of children. In order to provide recommandations to policy-makers to define priorities of action, the autors evaluated the coverage of growth monitoring and nutrition promotion in Rwanda and investigated associated factors. Awareness of the existence of growth monitoring and nutrition promotion appear to be one key issue to adress. The representativity of the studied sample is a strengh of the study.
  • This study described correlation between socio-demographic parameters and participation in growth monitoring and nutrition promotion sessions. It did not evaluate availability of these sessions in the area, which can also impact attendance. 

Author Response

Increasing coverage of growth monitoring and nutrition promotion among under-5 children can be a tool to improve nutritional and physical health of children. In order to provide recommandations to policy-makers to define priorities of action, the autors evaluated the coverage of growth monitoring and nutrition promotion in Rwanda and investigated associated factors. Awareness of the existence of growth monitoring and nutrition promotion appear to be one key issue to adress. The representativity of the studied sample is a strengh of the study.

This study described correlation between socio-demographic parameters and participation in growth monitoring and nutrition promotion sessions. It did not evaluate availability of these sessions in the area, which can also impact attendance. 

Response: Thank you very much for taking your time to read through our manuscript and providing us with very helpful comments. We have included as a limitation to the study, our inability to evaluate the extent of availability of growth monitoring and health promotion sessions in the study area, which can also impact attendance. This was due to the use of secondary data which did not capture information on the availability of growth monitoring and health promotion sessions.

Reviewer 3 Report

This is a very interesting article in which Authors wanted to investigate the risk factors related to the lack of growth monitoring. In my opinion the manuscript appears well performed and discussed. I don't have much quarrel with the overall analytics strategies of the paper, which I thought was adequate and sufficient. However, there are some issues with the style of the paper that should be addressed

Minor comments:

The introduction is clear but it should be more concise, the reader feels lost throughout. I suggest that it would be better to further discuss the malnutrition issue and to better define it.

Results: In the tables I struggle to understand the p value referring ? There are more than two groups, it would be better to clarify this aspect in the footnotes. 

"Table 1 show the weighted prevalence " ..please correct with shows.

Author Response

This is a very interesting article in which Authors wanted to investigate the risk factors related to the lack of growth monitoring. In my opinion the manuscript appears well performed and discussed. I don't have much quarrel with the overall analytics strategies of the paper, which I thought was adequate and sufficient. However, there are some issues with the style of the paper that should be addressed

Minor comments:

The introduction is clear but it should be more concise, the reader feels lost throughout. I suggest that it would be better to further discuss the malnutrition issue and to better define it.

Results: In the tables I struggle to understand the p value referring ? There are more than two groups, it would be better to clarify this aspect in the footnotes. 

"Table 1 show the weighted prevalence " ..please correct with shows.

Response: Thank you for the insightful comments. We have revised the manuscript in line with all comments provided. We are so grateful for your valuable time and efforts in reviewing the manuscript.